# Mitochondrial Abnormalities in Induced Pluripotent Stem Cells-Derived Motor Neurons from Patients with Riboflavin Transporter Deficiency

**DOI:** 10.3390/antiox9121252

**Published:** 2020-12-09

**Authors:** Fiorella Colasuonno, Enrico Bertini, Marco Tartaglia, Claudia Compagnucci, Sandra Moreno

**Affiliations:** 1Department of Science, LIME, University of Roma Tre, 00146 Rome, Italy; fiorella.colasuonno@uniroma3.it; 2Genetics and Rare Diseases Research Division, IRCCS Ospedale Pediatrico Bambino Gesù, 00146 Rome, Italy; marco.tartaglia@opbg.net; 3Unit of Neuromuscular and Neurodegenerative Diseases, Laboratory of Molecular Medicine, Department of Neuroscience, IRCCS Ospedale Pediatrico Bambino Gesù, 00146 Rome, Italy; enricosilvio.bertini@opbg.net

**Keywords:** riboflavin transporter deficiency, motor neurons, mitochondria, energy metabolism, electron microscopy, antioxidants, SOD2, oxidative stress, neurodegeneration

## Abstract

Riboflavin transporter deficiency (RTD) is a childhood-onset neurodegenerative disorder characterized by sensorineural deafness and motor neuron degeneration. Since riboflavin plays key functions in biological oxidation-reduction reactions, energy metabolism pathways involving flavoproteins are affected in RTD. We recently generated induced pluripotent stem cell (iPSC) lines from affected individuals as an in vitro model of the disease and documented mitochondrial impairment in these cells, dramatically impacting cell redox status. This work extends our study to motor neurons (MNs), i.e., the cell type most affected in patients with RTD. Altered intracellular distribution of mitochondria was detected by confocal microscopic analysis (following immunofluorescence for superoxide dismutase 2 (SOD2), as a dual mitochondrial and antioxidant marker), and βIII-Tubulin, as a neuronal marker. We demonstrate significantly lower SOD2 levels in RTD MNs, as compared to their healthy counterparts. Mitochondrial ultrastructural abnormalities were also assessed by focused ion beam/scanning electron microscopy. Moreover, we investigated the effects of combination treatment using riboflavin and N-acetylcysteine, which is a widely employed antioxidant. Overall, our findings further support the potential of patient-specific RTD models and provide evidence of mitochondrial alterations in RTD-related iPSC-derived MNs—emphasizing oxidative stress involvement in this rare disease. We also provide new clues for possible therapeutic strategies aimed at correcting mitochondrial defects, based on the use of antioxidants.

## 1. Introduction

Riboflavin transporter deficiency (RTD), formerly known as Brown-Vialetto-Van Laere syndrome, is a rare autosomal-recessive motor neuron disease. Typical clinical features include peripheral and cranial neuropathy, muscle weakness, sensory loss, diaphragmatic paralysis, and respiratory insufficiency, and multiple cranial nerve deficits, such as sensorineural hearing loss, bulbar symptoms, and loss of vision, due to optic atrophy [1,2]. Children affected with RTD show different phenotypes with variable prognosis, often influenced by the initiation of riboflavin (RF) treatment. In fact, some young patients treated with high-dose-RF supplementation show moderately improved muscle strength, motor function, respiration, hearing, and vision [3,4,5]. Novel approaches based on combined RF/antioxidant supplementation are likely to be even more efficacious, since symptoms are only partially reversed by the sole RF treatment.

In the last decade, mutations in human RF transporter genes *SLC52A2* (encoding RFVT2) and *SLC52A3* (encoding RFVT3) were demonstrated as causative factors for RTD [5,6]. RF represents an indispensable nutrient for human health, and its reduced intracellular availability compromises several vital processes, impacting the control of energy balance, particularly those depending on a proper redox status [7,8]. Mitochondria, major organelles responsible for lipid and reactive oxygen species (ROS) metabolism, are therefore likely to be impaired in their integrity and bioenergetics in RTD syndrome. 

Since in vivo models recapitulating symptoms and progression of RTD are so far lacking, we recently took advantage of induced pluripotent stem cell (iPSC) technology to reproduce the pathology in a patient-specific cellular model. We generated iPSCs from fibroblasts obtained from skin biopsies of patients carrying different mutations in the *SLC52A2* gene to characterize their phenotype from a morpho-functional viewpoint. Specifically, we documented mitochondrial abnormalities, involving shape, number, and intracellular distribution. Also, we demonstrated redox imbalance, resulting from the overproduction of superoxide anion, accompanied by an abnormal mitochondrial polarization state. Moreover, patients’ iPSCs showed altered expression of ROS-scavenging systems [9], encouraging the use of antioxidants for RTD treatment.

These results on patient-specific iPSCs, providing mechanistic insights into the disease pathogenesis to be used as a basis for innovative therapeutic approaches, prompted us to extend our study to motor neurons (MNs), i.e., the cell type most affected in patients with RTD. In the present investigation, we focus on mitochondrial morpho-functional features, studying their intracellular distribution and ultrastructural features, by confocal microscopy and focused ion beam/scanning electron microscopy, respectively. We performed immunofluorescence analysis of superoxide dismutase 2 (SOD2), as a dual mitochondrial and antioxidant marker, and of βIII-Tubulin, as a neuronal marker, in combination with an ultrastructural examination of the organelles, to shed light onto the role of oxidative stress in this neurodegenerative pathology. Moreover, we investigated the effects of a combined treatment employing RF and the antioxidant molecule N-acetylcysteine (NAC), to explore possible amelioration of neuronal phenotype associated with RTD involving mitochondrial compartment.

## 2. Materials and Methods

### 2.1. Derivation of iPSCs and Differentiation into MNs

Human iPSCs were obtained from healthy subjects and RTD patients, carrying *SLC52A2* gene mutations, displaying respiratory compromise, bilateral sensorineural hearing loss, reduced visual acuity, and progressive shoulder and axial muscle weakness. Skin fibroblasts were reprogrammed, by non-integrating episomal technology (Epi Episomal iPSC Reprogramming Kit, A15960, ThermoFisher Scientific, Waltham, MA, USA), using nucleofection. As oriP/EBNA1 vectors, these episomes contain five reprogramming factors (Oct4, Sox2, Lin28, Klf4, and L-Myc) and replicate extra-chromosomally only once per cell cycle. At this replication rate, the episomes are lost at a rate of approximately 5% per cell generation. The identified mutations were confirmed by Sanger sequencing, as previously reported [9]. Healthy and RTD iPSCs were then successfully differentiated into MNs, according to Corti and coll. [10]. Specifically, cells were maintained for 10 days in NeuroCult NS-A Basal Medium, Human (05750, StemCell Technologies, Vancouver, Canada), and incubated at 37 °C, with 5% CO_2_. From day 11, [0.1 μM] retinoic acid was added to the medium, which was changed every day. NeuroCult has also supplemented with [2 μM] Dorsomorphin and [3 ng/mL] Activin A, from day 17. Finally, [10 ng/mL] BDNF, [2 ng/mL] GNDF, [400 μM], dbcAMP and [200 μM] ascorbic acid, were added to cell medium from day 24. 

### 2.2. Immunofluorescence and Confocal Microscopy

For confocal analysis, cells underwent our previously described double immunofluorescence protocol [9], using the following primary antibodies: (i) Rabbit polyclonal antibody against mitochondrial superoxide dismutase 2 (anti-SOD2, ab13533, Abcam); (ii) mouse monoclonal antibody to βIII-Tubulin (T8578, Sigma-Aldrich); and (iii) rabbit polyclonal antibody against Neurofilament 200 (or SMI-32, N4142, Sigma-Aldrich). Cells were then incubated with the appropriate secondary antibodies, conjugated with Alexa Fluor 488 or Alexa Fluor 594 (Invitrogen, Carlsbad, CA, USA). Nuclei were stained with 1 μg/mL Hoechst (33342, Invitrogen), and slides were observed in a Leica TCS SP5 confocal microscope (Leica, Wetzlar, Germany). Random images for statistical analysis purposes were captured by Leica Application Suite software, and representative images were composed in an Adobe Photoshop CS6 format (Adobe Systems Inc., San Jose, CA, USA).

### 2.3. Electron Microscopy

Ctrl and RTD MNs were plated in a chamber slide (Lab-Tek II Chamber Slide System, ThermoFisher Scientific). Some samples were treated with 1 µM RF (R9504, Merck KGaA, Darmstadt, Germany) and 100 µM N-acetylcysteine amide (NAC) (A0737, Merck KGaA) overnight.

Treated and untreated cells were fixed and embedded in epoxy resin, according to Colasuonno and coll. [11]. Electron microscopic observations and analyses were performed by a DualBeam focused ion beam/scanning electron microscopy (FIB/SEM, Helios Nanolab, FEI, Hillsboro, OR, USA). Random images for statistical analysis purposes were electronically captured. Representative images were composed in an Adobe Photoshop CS6 format.

### 2.4. Statistical Analysis

ImageJ (NIH) software was used to quantify immunofluorescence intensity, while statistical analysis was performed using Prism software (GraphPad Software, Inc., La Jolla, CA, USA). A minimum of 300 cells/sample per experiment, were analyzed for fluorescence intensity levels, then a parametric t-test was used, since a normal distribution was assumed.

For mitochondrial quantitative analyses, three samples/cell culture type and 10 cells/sample per experiment, were examined, by manually counting regular and altered mitochondria of Ctrl and RTD MNs, and their percentage was calculated. Two-way ANOVA was used to analyze differences among genetic conditions and treatments, followed by a Bonferroni post-hoc test. 

The results are presented as means ± SD of n ≥ 3 independent experiments. A *p*-value of 0.05 or less was considered statistically significant (*), while ** indicate *p*-value equal to or lower than 0.01.

## 3. Results

### 3.1. Impaired SOD2 Distribution and Levels Are Associated with Altered Morphology of RTD Neurons

Confocal analyses of RTD and Ctrl iPSCs-derived MNs were performed after immunofluorescence, using SOD2 as a dual marker for mitochondrial number and for an antioxidant response, and βIII-Tubulin (βIII-TUB) as a neuronal differentiation marker. While the distribution of βIII-TUB in Ctrl cells highlights long neuritic processes, forming an intricate network, in RTD MNs, this neuronal marker reveals the presence of shorter neurites (Figure 1, red signal). Immunofluorescence analysis using neurofilament antibody (SMI-32), as a specific neuronal marker, shows similar morphological differences between Ctrl and RTD MNs (Appendix A). Such morphological changes are accompanied by reduced immunoreactivity to the mitochondrial marker SOD2 (Figure 1, green signal in confocal images). Accordingly, quantitative analysis demonstrates significantly lower SOD2 expression levels in RTD MNs, as compared to their healthy counterpart (Figure 1e).

### 3.2. Aberrant Mitochondrial Ultrastructure in RTD Neurons Is Rescued by RF/NAC Treatment

Data from confocal analyses prompted us to investigate mitochondrial features, as altered enzyme content likely reflects morphological changes. Indeed, FIB/SEM analyses of differentiated MNs highlights profound mitochondrial abnormalities, associated with RTD phenotype. These organelles appear swollen, with disrupted *cristae* (Figure 2b,d), as opposed to mitochondria of Ctrl MNs, showing regular morphology (Figure 2a,c). Statistical analysis to evaluate the occurrence of altered mitochondria demonstrates the presence of a significantly higher percentage of damaged mitochondria, over their total number, in RTD vs. Ctrl MNs (Figure 2i). No significant differences in the number of total mitochondria were observed between the conditions (data not shown). Moreover, while Ctrl cells displayed several lysosomes and autophagosomes, fewer of these organelles were present in RTD MNs.

We also investigated at the ultrastructural level whether N-acetylcysteine (NAC), a glutathione (GSH) precursor, could exert a protective effect when administered with RF. As shown by electron microscopic images (Figure 2e–h), this combined treatment restores normal mitochondrial morphology in RTD MNs. In fact, statistical analysis demonstrates a significant decrease (** *p* ≤ 0.01) of the number of damaged mitochondria in patients’ MNs after RF + NAC treatment (Figure 2i).

## 4. Discussion

Riboflavin (RF) and its derivatives, FMN and FAD, play a crucial role in essential cellular processes, including mitochondrial energy metabolism, stress responses, vitamin, and cofactor biogenesis, ensuring the catalytic activity and folding/stability of flavoenzymes [7,12]. RF is, therefore, an indispensable nutrient for human health and represents one of the neglected antioxidants that may have an action independently or as a component of the glutathione redox cycle [13]. RF deficiency impacts redox balance, compromising energy metabolism pathways, and antioxidant defense mechanisms [7,13]. Insufficient availability of the vitamin results in severe clinical conditions, particularly affecting MNs; thus, sharing traits with amyotrophic lateral sclerosis (ALS) [14]. Emphasizing the importance of RF in human physiology, and furthermore, its efficient absorption and homeostasis are RTDs caused by recessive, biallelic mutations in the genes encoding human RF transporters [1]. Several in vivo and in vitro models have been generated to dissect the molecular mechanisms underlying RF deficiency [15,16,17]; however, a model recapitulating the human pathology is still lacking. Thus, we recently developed a patient-specific induced pluripotent stem cell (iPSC) model to study RTD phenotype from a morpho-functional viewpoint [9,18].

In the present study, we took advantage of iPSC potential, to differentiate MNs. Hypothesizing the involvement of oxidative stress of mitochondrial origin in RTD MNs, we investigated whether the same alterations demonstrated in iPSCs derived from RTD patients [9], were maintained when undergoing differentiation protocol. This issue seemed crucial, given possible therapies targeting oxidative stress, particularly mitochondrial function, in RTD patients.

Overall, our data on patient-specific MNs emphasize the involvement of mitochondrial alterations in RTD pathogenesis (Figure 3). Indeed, significantly lower SOD2 immunofluorescence levels were detected in RTD vs. normal cells, revealing impaired mitochondrial functionality, with special reference to antioxidant properties. Notably, specific modulation of SOD2 in RTD condition is consistent with the reported role for RF, as an effective agent to boost the activity of this enzyme [19]. Such abnormalities in O_2_^−.^ Scavenging ability, in total agreement with data obtained in undifferentiated cells from the same patients [9], likely result in exacerbated redox imbalance. This is, however, thought to be mainly contributed by defective energy metabolism pathways controlled by flavoproteins [7,8]. Altered levels of antioxidant systems were also found in experimental dietary RF deficiency, supporting pleiotropic roles for RF in regulating redox balance, including gene expression regulation [15,16,17,20,21,22,23]. Whatever its origin, oxidative stress may well account for the relatively poor development of neurites in RTD MNs, as revealed by βIII-TUB immunostaining. Patient-specific MNs display, in fact, shorter and fewer processes, as compared to their healthy counterparts, as suggested by a previous study [24]. 

Ultrastructural results correlate well with confocal data. Indeed, FIB/SEM analysis shows in RTD MNs remarkable mitochondrial dysmorphologies, strongly suggesting their dysfunction [9,11]. Altered features associated with RTD condition include disrupted mitochondrial *cristae* and the presence of vacuoles in the mitochondrial matrix. Since mitochondrial dysfunction is known to trigger mitophagy, as a quality control mechanism, we looked for autophagic vacuoles. However, RTD cells failed to show an appreciable number of double-membrane limited vacuoles, containing intact or partially digested organelles. Differently, in healthy MNs, we detected numerous autophagosomes and lysosomes, suggesting active autophagic processes, as expected for an intensely active cell type. Possible involvement of insufficient/defective systems of repair/removal of damaged organelles in RTD pathogenesis has been suggested [8,24], and certainly deserves further studies. However, our present and previous data still support mitochondrial-generated redox imbalance as responsible for the RTD altered phenotype. 

We also explored novel therapeutic approaches by combining RF supplementation—known to partially ameliorate RTD phenotype—with the antioxidant, namely, N-acetylcysteine (NAC), which is widely employed for free radical scavenging and glutathione replenishment. Indeed, this combined treatment proved successful, in improving patients’ cell morphology, particularly as regards mitochondrial ultrastructural features. This result is consistent with work in progress at our laboratories showing that such combined treatment results in increased neurite length in RTD MNs [25], possibly by the restoration of ROS-sensitive microtubule dynamics [26,27]. Studies on different pathological conditions have associated the efficacy of NAC with its action in cystine-glutamate exchange, glial glutamate transporters normalization, and in restoring glutamatergic tone on presynaptic receptors in reward regions of the brain [28,29]. Importantly, NAC has a long-established safety record and does not require titration to achieve the target dose [28]; thus, proposing this compound for future therapeutic studies aimed at improving clinical outcomes of RTD patients. 

## 5. Conclusions

Human iPSCs represent an innovative model for numerous clinical studies, opening new perspectives to research and applications in the field of rare neurodegenerative disorders, which are generally difficult to study, due to the lack of models recapitulating onset and progression of the pathology. In this context, our recent [9] and present data lay the foundations for using iPSCs as in vitro model for a rare pediatric disease, whose devastating symptoms urgently require the development of novel therapeutic strategies. Specifically, iPSCs generated by RTD patients’ fibroblasts and then differentiated into MNs offer us the opportunity to explore a drug treatment, identifying a combination of antioxidant molecules, able to improve morpho-functional features of patients’ cells. While supporting the innovative potential of our iPSC model of RTD, our work demonstrates an impaired mitochondrial ROS scavenging system, accompanied by morphological alteration of the organelles in RTD MNs. We also demonstrate the effects of combined treatment with RF and NAC on the mitochondrial morphology of RTD MNs. Such beneficial effects of antioxidants emphasize the transactional aspect of the present study, stimulating future investigation aimed at identifying the specific pathways enhanced by the molecules, involving the mitochondrial compartment.

## Figures and Tables

**Figure 1 antioxidants-09-01252-f001:**
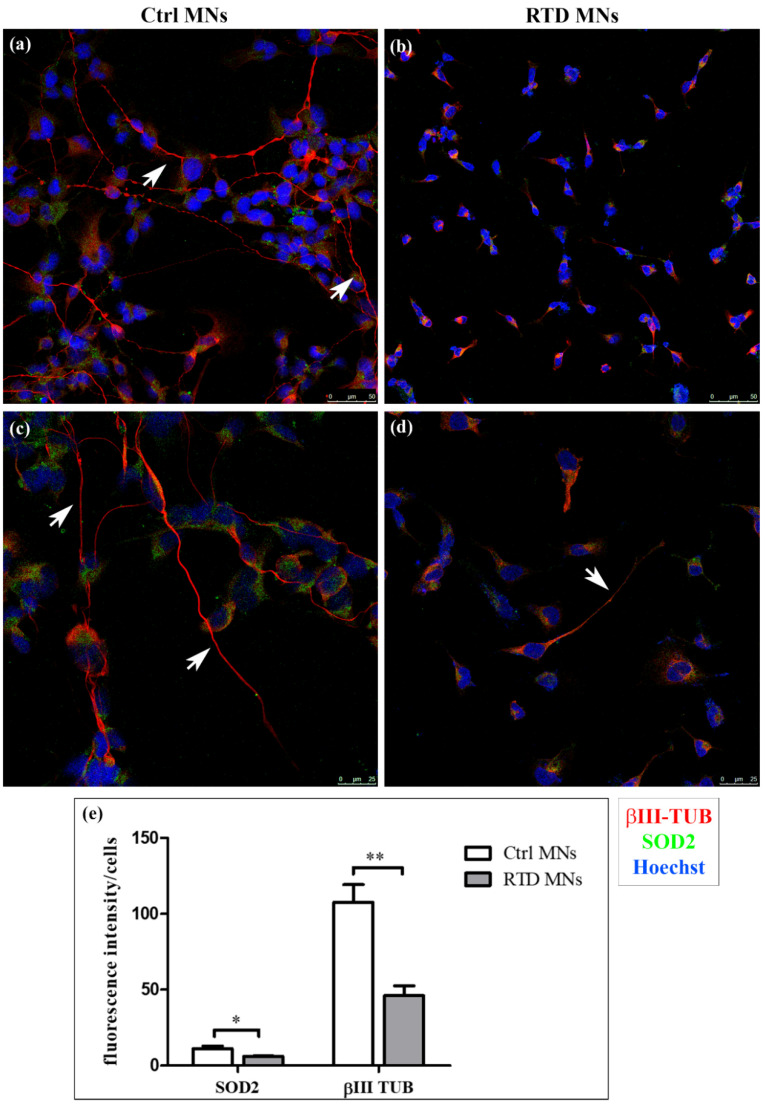
Confocal analysis of riboflavin transporter deficiency (RTD) vs. Ctrl motor neurons (MNs). (**a**–**d**) Immunofluorescence images showing the distribution of βIII-Tubulin (βIII-TUB) (red) and superoxide dismutase 2 (SOD2) (green). Nuclei are stained with Hoechst (blue). White arrows indicate neurites in (**a**,**c**–**e**) Significantly lower levels of βIII-TUB and of the mitochondrial marker SOD2 are detected in diseased MNs (* *p* ≤ 0.05 vs. Ctrl MNs; ** *p* ≤ 0.01 vs. Ctrl MNs). White columns indicate Ctrl cells, while grey ones indicate diseased MNs.

**Figure 2 antioxidants-09-01252-f002:**
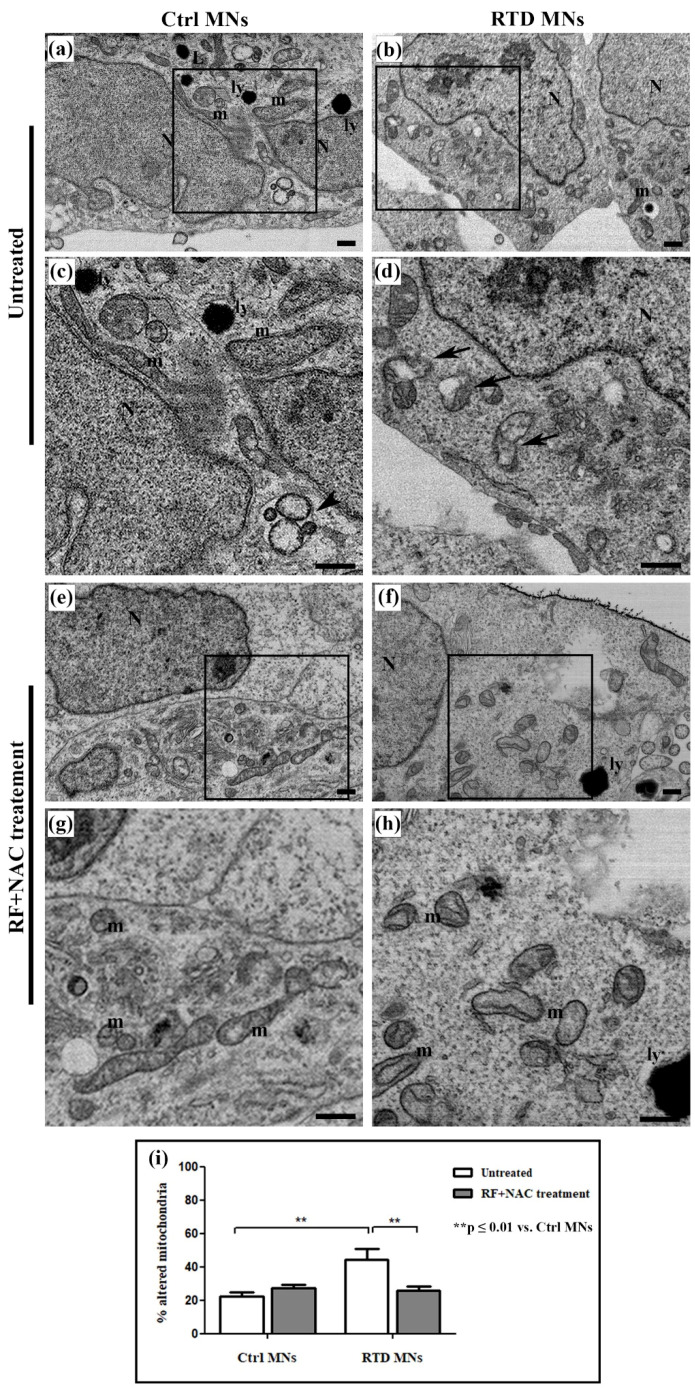
Ultrastructural analysis of RTD vs. Ctrl MNs, before and after riboflavin (RF) + N-acetylcysteine (NAC) treatment. (**a**–**d**) FIB/SEM micrographs show the presence of several altered mitochondria in RTD MNs, with swollen morphology and deranged cristae (**a**,**c**), as opposed to regularly shaped organelles in Ctrl cells (**b**,**d**). After RF + NAC treatment (**e**–**h**), improved mitochondrial ultrastructure in RTD MNs is observed. (**i**) Statistical analysis demonstrates a significantly higher number of damaged mitochondria in RTD MNs as compared to Ctrl. RF+NAC treatment restores mitochondrial morphology in RTD MNs (** *p* ≤ 0.01 vs. Ctrl MNs). Scale bars, 1 μm. N, nuclei; m, regular mitochondria; ly, lysosomes; black arrows, altered mitochondria; black arrowheads, autophagosomes.

**Figure 3 antioxidants-09-01252-f003:**
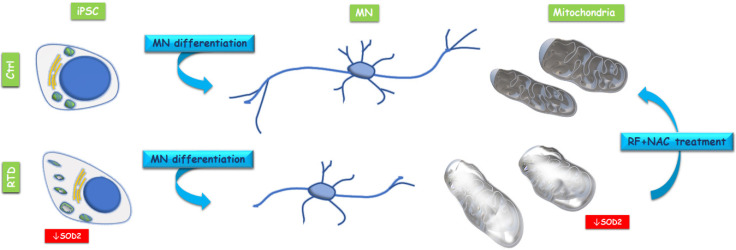
Conceptual schema showing mitochondrial alteration in RTD induced pluripotent stem cells (iPSCs) and iPSC derived MNs and beneficial effects of RF + NAC treatment on mitochondrial morphology.

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
