# Peer review of "Mitochondrial Abnormalities in Induced Pluripotent Stem Cells-Derived Motor Neurons from Patients with Riboflavin Transporter Deficiency"

_antioxidants, 2020, doi:10.3390/antiox9121252_

Round 1

Reviewer 1 Report

Comments to the authors:

The paper by Colasuonno Fiorella, et al., presents some interesting findings and a unique protocol to generated iPSC lines from affected individuals as an in vitro model of the RTD disease and documented mitochondrial impairment. In the present work, the authors extend the studies to motor neurons, which is mostly affected in patients with RTD. However, in the results and methods sections, some issues need to be clarified.

Major comments

Material and Methods. 

Derivation of iPSCs and differentiation into MNs

The authors briefly describe this methodology and present two references in this section. However, the authors must describe some individuals features (age, gentle and clinical symptoms of the patients) in this manuscript.

Since the methodology of the differentiation protocol is unique, the authors must briefly describe the protocol of the cell differentiation of the isolated iPSCs–derived MNs.

Results

III beta-tubulin is predominantly in neurons and testis. The authors need to provide or discuss the evidence that they culture iPSC-derived MNs. Authors must present other neuronal markers that they used in the present manuscript. 

In Figure 1, Are there more cells in the control image?. The authors use  BIII-tubulin as a neuronal marker. Can the authors point out the different parts of the MN in figure 1?. The cells can develop axon/dendrites from the isolated CTL iPSCs–derived MNs.?  Are unable to from axon/dendrites in the isolated RTD iPSCs-derived MNs?. What will have happened if the presence of RF+NAC treatment with cellular morphology in RTD iPSCs-derived MNs?. Do they form axon/dendrites? The authors need to discuss these questions in the discussion sections.

The authors present the results using Confocal analysis and electron microscopy. However, I suggest confirming these results using a different technique such as western blot for SOD2  (figure 1) to fully demonstrate the impaired mitochondrial functionality as the authors mention in the discussion section lines 70 and 72.

Lines 31 and 32. The Authors describe the number of damaged mitochondria in RTD vs. Ctrl MNs (Fig 2i). How they measure this parameter? Do the authors count the number of the damaged mitochondria / total number of mitochondria in the image?. Is the number of total mitochondria different between RTD and CTL?   Do the mitochondrial change its cellular position? The authors need to clarify these questions.

The authors mentioned the beneficial effects of RF+NAC treatment. Do the ROS signaling pathway is involved in axon and dendrites´ formation?

Figures

Figure 1E. The authors need to describe the figure and the results added in the results section. For example, describe the graphics and the columns.

Figure 3. Is it not clears the blue titles, Can the authors use another color?

Reviewer 2 Report

The authors produced a very interesting paper regarding "Mitochondrial Abnormalities in iPSC-derived Motor Neurons from Patients with Riboflavin Transporter Deficiency". I consider the manuscript very interesting but, before it could be considered suitable for publication, I suggest the following revisions:

  • The statistical analysis section lacks of the exact statistical test used, and the authors have to add and describe it.
  • Typos and English language corrections have to be correct.

Update (2020.11.19):

  • The statistical analysis section lacks of the exact statistical test used, and the authors have to add and describe it. For example, what test was applied with PRIMS? ANOVA? Parametric? Non-Parametric? Moreover, the used test has to present a post-hoc test, to ensure the reliability of statistical results, such as Tukey post-hoc, False Discovery Rate or Bonferroni corrections. It is fundamental to detail the used statistical approach to ensure the quality of outcomes.
  • Typos and English language corrections have to be correct. I suggest the authors to request the help of a mother-tongue to obtain the best result.

Author Response

  1. The statistical analysis section lacks the exact statistical test used, and the authors have to add and describe it. For example, what test was applied with PRIMS? ANOVA? Parametric? Non-Parametric? Moreover, the used test has to present a post-hoc test, to ensure the reliability of statistical results, such as Tukey post-hoc, False Discovery Rate or Bonferroni corrections. It is fundamental to detail the used statistical approach to ensure the quality of outcomes. We thank the reviewer for this point. In the revised version of the manuscript we added statistical test used. Specifically, we used a parametric t test for immunofluorescence analysis of β-tubulin III and SOD2 markers. We then used a two-way ANOVA followed by Bonferroni post-hoc test to analyze the number of damaged organelles.
  2. Typos and English language corrections have to be correct. I suggest the authors to request the help of a mother-tongue to obtain the best result.
    We thank the reviewer for the suggestion. We have checked the English language with the help of a mother-tongue.

Round 2

Reviewer 2 Report

The authors addressed all suggested points.